# *Bis*(Tryptophan) Amphiphiles Form Ion Conducting Pores and Enhance Antimicrobial Activity against Resistant Bacteria

**DOI:** 10.3390/antibiotics10111391

**Published:** 2021-11-12

**Authors:** Mohit Patel, Saeedeh Negin, Joseph Meisel, Shanheng Yin, Michael Gokel, Hannah Gill, George Gokel

**Affiliations:** 1Department of Chemistry & Biochemistry, University of Missouri, St. Louis, MO 63121, USA; mohitpatel87@gmail.com (M.P.); saeedehnegin@gmail.com (S.N.); syx97@mail.umsl.edu (S.Y.); gokel.mike@gmail.com (M.G.); hannahgill122@gmail.com (H.G.); 2Department of Biology, University of Missouri, St. Louis, MO 63121, USA; 3Department of Chemistry, George Washington University, Washington, DC 20052, USA; jmeisel@gwu.edu

**Keywords:** adjuvant, amino acids, amphiphile, antimicrobial, bacteria, efflux pump, ion channel, pore formation, resistant bacteria, tryptophan

## Abstract

The compounds referred to as *bis*(tryptophan)s (BTs) have shown activity as antimicrobials. The hypothesis that the activity of these novel amphiphiles results from insertion in bilayer membranes and transport of cations is supported by planar bilayer voltage-clamp studies reported herein. In addition, fluorescence studies of propidium iodide penetration of vital bacteria confirmed enhanced permeability. It was also found that BTs having either meta-phenylene or n-dodecylene linkers function as effective adjuvants to enhance the properties of FDA-approved antimicrobials against organisms such as *S. aureus*. In one example, a BT-mediated synergistic effect enhanced the potency of norfloxacin against *S. aureus* by 128-fold. In order to determine if related compounds in which tryptophan was replaced by other common amino acids (H_2_N-Aaa-linker-Aaa-NH_2_) we active, a family of analogs have been prepared, characterized, and tested as controls for both antimicrobial activity and as adjuvants for other antimicrobials against both Gram-negative and Gram-positive bacteria. The most active of the compounds surveyed remain the bis(tryptophan) derivatives.

## 1. Introduction

Numerous reports have appeared warning of the increasing threat of illnesses caused by antibiotic-resistant bacteria [1]. Resistance has developed by at least some organisms to nearly all known antimicrobial agents [2]. It is now generally recognized that new and effective solutions to deal with this crisis are essential [3,4] and overdue [5]. Although a few novel antibiotics such as teixobactin [6] have emerged recently, most new antibiotics are derivatives of existing classes of antimicrobials. The crisis is of such significance that both the World Health Organization (WHO) and the Centers for Disease Control and Prevention (CDC) [1] have issued recent, extensive reports. In 2017, the WHO listed 12 bacterial strains that are the greatest threat to public health [7]. It was asserted in that report that the current rate of antibiotic development is inadequate to address this crisis. Of particular concern are multidrug-resistant (MDR) Gram-negative bacteria and methicillin-resistant *Staphylococcus aureus* (MRSA), which constitute an urgent threat.

Amphiphilic molecules, both natural and synthetic, are known to manifest antimicrobial properties [8]. Amphiphiles localize in and disrupt cell membranes in bacteria. Natural amphiphiles [9] (often peptides [10]) such as colistin [11] and daptomycin are highly effective at inhibiting bacterial growth. Their use has generally been limited owing to toxicity, but the rise in antibiotic resistance has increased demand for them [12]. Colistin [13,14] has been in clinical use for over 50 years, but resistance to it has recently been reported in the U.S. [15,16,17,18]. Recent research has focused on using colistin as an adjuvant [19,20,21]. The rate of resistance development to such antibiotics as tetracycline usually allows a 10–15 year lifetime for a new antibiotic in clinical use [5]. Although amphiphiles generally acquire resistance slowly, they are sometimes associated with renal [22] and cardiac cytotoxicity. Recent research involving colistin emphasizes changes to the hydrophobic tail or alterations in the number of positive charges, but significant success has proved to be elusive. Colistin is also now used in prodrug form as colistin methanesulfonate in an effort to lower its cytotoxicity [23].

Previous studies in our laboratory have produced the families of compounds we call lariat ethers [24,25,26] and hydraphiles [27]. The lariat ethers were designed originally as crown ethers having attached donor group-containing side arms. These side chains provide a ring-bound cation with apical solvation [28]. The lariat ethers proved to be excellent ion carriers [29]. When the donor groups were eliminated from the side arms, certain alkyl lariats aggregated to form channels that function in phospholipid bilayers [30]. Of course, they may continue to function as carriers as well. The hydraphiles were designed as channels that would be active in bilayer membranes and their function as such has been fully characterized and reported [31]. Both classes of compounds exhibit antimicrobial properties and both classes of compounds function as potency-enhancing adjuvants for common antibacterials to which organisms have become resistant [32,33].

Given our success in developing amphiphiles that exhibited antimicrobial properties, we sought to develop other classes of compounds able to insert in membranes that may exhibit antimicrobial properties. The *bis*(tryptophan)s (BTs) [34] that are the subject of this report were designed as potential amphiphiles presuming that the amino acid’s indoles would function as membrane anchors [35,36]. It is well known that tryptophan occurs in many peptides and proteins, such as gramicidin [37,38] and the KcsA voltage-gated ion channel, [39] only at the membrane boundaries. Our own previous work showed that indoles substituted by alkyl chains at either the *N*- or 3-position could form stable liposomes [40]. We therefore prepared a family of compounds in which we expected tryptophan to function as a head group. Our design schematic had the general formula H_2_N-Trp-Y-Trp-NH_2_, in which Y was alkyl or aryl. The expectation was that tryptophan’s indole would serve as a membrane anchor and the molecule would enhance the permeability of the membrane in which it was resident. We have previously reported that several of the *bis*(tryptophan) compounds exhibited antimicrobial potency and low cytotoxicity [24].

The *bis*(tryptophan)s recovered tetracycline activity against a tetA efflux pump expressing tetracycline-*resistant Escherichia coli* (tet^R^) [34]. Limited structure-activity studies for these tryptophan derivatives revealed that charged ammonium moieties, an appropriate spacer, and the presence of tryptophan as the terminal amino acids were all critical for antimicrobial activity. In one example for which d- and l-isomers were prepared, both proved to be active. In this case, the compound having the d,d-configuration was found to be more active than the otherwise identical l,l-derivative against a strain of *E. coli*. Several of the *bis*(tryptophan)s showed greater potency against *Staphylococcus aureus* (ATCC 29213) than *E. coli*. We now report additional antimicrobial studies, the membrane activity of several BTs, insight into the mechanism of action of these *bis*(tryptophan)s, and a group of new *bis*(amino acid)s.

## 2. Results and Discussion

Two key questions persisted in our study of *bis*(tryptophan)s. First, our hypothesis was that these amphiphiles insert into bacterial membranes (boundary layers) and both increase permeability and affect such enzymes as efflux pumps. Could evidence be developed to support either or both of these assumptions? Second, was the formation of membrane-active amphiphiles based on amino acid head groups possible with any of the other common amino acids? We attempted to resolve the first question by using the planar bilayer voltage clamp experiment and fluorescence techniques. The second issue was addressed by preparing derivatives of the other common amino acids and determining their biological activity, if any. The initial group of tryptophan derivatives that were prepared [34] is shown in Figure 1.

### 2.1. Channel and Biological Activity

As noted in the introduction, the tryptophan derivatives were prepared in the expectation that this particular amino acid would serve as a membrane anchor for a membrane-active amphiphile [34]. We recognized that the spacer chain lengths of the compounds prepared would be insufficient for these amphiphiles to span typical bilayer membranes [41]. However, membrane interactions are still possible and a surface-stabilized partial insertion into a membrane is certainly plausible [42]. The planar bilayer voltage clamp (BLM) experiments [43] reported below are useful to detect membrane insertion and especially controlled ion transport.

The BLM experiment was used to assess the membrane behavior of compound **1** (HCl·H_2_N-L-Trp-*m*-C_6_H_4_-L-Trp-NH_2_·HCl, W-*m*C_6_H_4_-W). As recorded in Table 1 below, **1** was not the most biologically active structure under study. However, the greatest amount of data had been obtained for it, including comparisons with its structural isomers and stereoisomers. Molecular models suggested that the amino-to-amino span in **1** is ~12 Å. If the tryptophans are in both distal positions of an extended molecule, the overall span could exceed 20 Å. Even in that case, this length seems somewhat short to span a bilayer as a single molecule. The hydrocarbon regimes of bilayer membranes are usually estimated at 30–35 Å, although the insertion of channels may compress the inner and outer membrane leaflets so that the membrane thickness is diminished. Further, the amphiphile may form a barrel stave pore [42,44,45] or a torodial pore [42,46] in which one leaflet of the bilayer reorganizes to pass ions when the pore span is otherwise too short. The membrane behavior of W-*m*C_6_H_4_-W (**1**) was investigated in a planar soybean asolectin bilayer membrane.

A BLM study of W-*m*C_6_H_4_-W (**1**) produced inconsistent results. In contrast, however, the corresponding D,D-isomer (**2**) showed clear evidence for channel formation. A typical trace is shown in Figure 2. Classical open-close behavior was observed for **2** and the ion currents were generally in the expected range of ~30 picoSiemens. The trace shown indicates that there are two stable open states. This could correspond either to the detection of two open channels or to aggregate formation of undetermined monomer numbers. In either case, the regularity of the trace evinced by **2** is clear.

Since differences in the two stereogenic centers do not alter overall length, there may be a selective interaction of **2** with the chiral element, i.e., the *R* stereochemistry of the sn-2 center in asolectin’s phospholipid monomers. It seems more likely, however, that any difference results from experimental vicissitudes of the complex planar bilayer experiment.

Little membrane activity was anticipated for the *ortho*-phenylene analog (**3**) owing to the shorter span of the compound. The estimate of N↔N distance for either **1** or **2** is ~12 Å while it is ~9 Å for **3**. Of course, the overall length of these compounds is less consequential if a barrel stave or toroidal pore forms. Notwithstanding the distance differences, **3** showed the most regular open-close behavior that was observed for the phenylene compounds. A typical trace is shown in Figure 3.

Compound **6**, W-C_12_-W, was the most biologically active of the bis(tryptophans). A planar bilayer conductance trace is shown in Figure 4. During the experiment, it typically showed some indication of channel activity, but quickly revealed spiking behavior. An attempt was made to obtain planar bilayer results for **1**–**6**, but the traces shown for **2** and **3** (Figure 2 and Figure 3) were the only compounds that afforded reproducible conductance traces.

As was the case with the other diaminophenylene isomers, stable pore formation (BLM traces) was not observed for the *para*-phenylene BT [W-*p*C_6_H_4_-W (**4**)]. Of the three phenylenediamine derivatives, the *p*-isomer was the least potent against the bacteria tested. The small difference in span alone does not appear to explain either the low potency or the lack of detectable planar bilayer activity. As above, the latter may be an experimental issue. Perhaps the *para*-compound is conformationally more rigid than the *ortho*- or *meta*-isomers. The melting points for the *ortho*- (**3**), *meta*- (**1**) and *para*-isomeric (4) BTs are 201 °C, 154–156 °C, and 237 °C, respectively, suggesting a stronger crystal matrix that comports with lower solubility of the *para*-isomer.

The conductances observed for compounds **2** and **3**, which formed stable pores were appropriate (~30 pS) for transport of K^+^. It is currently unknown if the active compounds function as monomers or as aggregates. If aggregation does occur, it seems most likely with W-*o*C_6_H_4_-W (**3**), which is the shortest of the BTs. Aggregation may also occur with **2** and **6**, each of which shows more than one conductance state, possibly indicating pore formation by **2**_n_ and **6**_n_ assemblies [30].

### 2.2. Membrane Permeability

The formation of pores could significantly affect the amount of an exogenous compound that penetrates the target organism. The presence of pores should reorganize (disorganize) or disrupt the adjacent phospholipids and increase the membrane’s permeability [42]. Of course, membrane disrupting compounds that fail to form organized pore assemblies could also enhance membrane leakage. A bacterial permeability analysis was conducted that comprised a fluorescent assay of the three most bioactive BTs. *S. aureus* was exposed to propidium iodide over a range of concentrations in the presence or absence of BTs. Propidium iodide normally fails to penetrate bilayer or bacterial boundary membranes. The fluorescence arising from penetration of the cells was evaluated quantitatively to establish which BT engendered the greatest membrane permeability.

Compounds **1** (W-*m*C_6_H_4_-W), **2** (w-*m*C_6_H_4_-w) and **6** (W-*n*C_12–_W) were chosen for study. The data in Table 2 (presented later in this report) showed that W-*n*C_12_-W (**6**) was the most bioactive compound in the group surveyed, followed by w-*m*C_6_H_4_-w (**2**) and W-*m*C_6_H_4_-W (**1**). Both the d,d- and l,l-isomers (**1** and **2**) were studied as their antimicrobial potencies differed. *S. aureus* was the organism used in the study. In each case, it was exposed to propidium iodide (see Figure 5) in the presence of the indicated concentration of BT or control. Propidium iodide does not normally penetrate vital cells. When it does enter cells, it intercalates into DNA and a fluorescent signal is detected. The intensity of the fluorescence is proportional to the penetration of the salt. Controls in the study shown in Figure 6 included *S. aureus* alone, in the presence of 0.5% DMSO, and either 0.1% or 0.01% of Triton X-100 (abbreviated TX100) detergent. The percentages of Triton X-100 expressed as concentration correspond to 170 µM and 17 µM, respectively. The DMSO control is important to show that the cosolvent used has no effect on the system. Triton X-100 was used to demonstrate that a simple detergent at low concentration does not enhance permeability.

The concentration range studied for added BTs was 500 nM to 32 µM. The system was indifferent to the presence of 0.5% DMSO. Some increase in membrane permeability is apparent with Triton X-100 controls having concentrations up to 170 µM. The BTs showed no significant increase in fluorescence and thus propidium iodide penetration below 2 µM. However, from 4 µM to 32 µM, **6** (W-*n*C_12_-W) mediated propidium iodide fluorescence (cell penetration) by approximately threefold. While little change in fluorescence was observed for **1** (W-*m*C_6_H_4_-W), isomer **2** (w-*m*C_6_H_4_-w) fostered an increase in fluorescence to approximately 1.8-fold at 32 µM.

The MICs (Table 2 and Table 3) of W-*n*C_12_-W (**6**) against *S. aureus* and *E. coli* were 4 µM and 10 µM, respectively. The d,d- (**2**) and l,l-forms (**1**) of W-*m*C_6_H_4_-W were both active, but showed different biological potencies. Against *S. aureus*, the d,d-compound’s MIC was 16 µM compared to 128 µM for the l,l-form. Likewise, the d,d-compound’s MIC against *E. coli* was 28 µM compared to 48 µM for the l,l-isomer.

The results of the membrane permeability study correlate well with the MIC of W-C_12_-W (**6**) i.e., 4 µM. The first significant permeability enhancement over controls is observed at the same concentration. Successive doubling of the concentration affords a nearly linear increase in permeability as gauged by propidium iodide penetration. [Actually an exponential curve (R^2^ = 0.998) owing to the doubling of MICs (data not shown).] A similar profile is observed for d,d-Trp-*m*C_6_H_4_-Trp (**2**). The l,l-isomer showed little activity over this concentration range.

A surprising observation arising from the propidium iodide experiment concerns the difference in propidium iodide penetration of *S. aureus* in the presence of W-*m*C_6_H_4_-W (**1**) and its isomer, w-*m*C_6_H_4_-w (**2**). The latter was prepared as a test for metabolic stability. It was surmised that d-Trp linkages would be metabolized less readily than l-Trp by endogenous amidases and therefore be more persistent within the organism. There was no expectation that differences in stereochemistry would affect propidium iodide penetration through *S. aureus* membranes. The only stereogenic element in typical phospholipid bilayer membranes occurs at the sn-2 carbon of glycerol. Bittman, Koeppe, and coworkers showed that the channel-forming peptide gramicidin’s conductance did not depend on the phospholipid stereochemistry [47]. Further, the glyceryl regime of bilayers is not in contact with bulk water, but is in an environment estimated to have a dielectric constant near 25 [48]. Of course, the boundary layer/membrane in Gram-positive *S. aureus* is more complex than a simple bilayer. Even so, the difference in permeability behavior between these two compounds is intriguing although not currently understood.

### 2.3. Checkerboard Experiments

In previous studies with synthetic amphiphiles, we found that synergy with FDA-approved drugs was possible. Several candidates were available among compounds **1**–**6**, but combination experiments were conducted only with W-*m*C_6_H_4_-W (**1**), w-*m*C_6_H_4_-w (**2**) and W-*n*C_12_-W (**6**), owing to their superior potencies. Studies were conducted with the three BTs and with an efflux pump inhibitor control [carbonyl cyanide *m*-chlorophenyl hydrazine (CCCP)] in the presence of either norfloxacin or ethidium bromide. CCCP was used as a positive control. The target organism was *S. aureus* expressing a norA efflux pump. This efflux pump provides resistance to norfloxacin and ethidium bromide by actively pumping them out. Efflux pump function is dependent on a cation gradient. Non-rectifying channel activity should lead to disruption of ion gradients on which efflux pumps depend. Pore formation should affect ion gradients, which in turn will reduce pump function. This, in combination with membrane disruption, should allow for greater accumulation of substrate, here norfloxacin and ethidium bromide, in the bacterial cytoplasm. Enhanced antimicrobial efficacy or recovery of potency should be observed regardless of resistance mechanism. This hypothesis was tested by conducting the checkerboard experiments shown below.

The target organism was *S. aureus* 1199B. The data are presented in Figure 7 in the checkerboard format. The *y*-axis in each graph shows the concentration of BT or CCCP. In panels A, B, and C, the *x*-axis shows [norfloxacin]. In panels D, E, and F, the *x*-axis shows [ethidium bromide] (EthBr). Panels G and H parallel A-C and F, respectively, except that [CCCP] is shown on the *y*-axis. The intensity of the red color indicates the growth of viable bacteria. The darker the hue the greater the cell viability. The MIC concentrations of the compounds used in each experiment are marked. The growth/viability of *S. aureus* in the presence of the BT, norfloxacin, or ethidium bromide alone is also indicated by the shade of red. Ethidium bromide is not an antibiotic, but it is used here to screen the efficacy of efflux pumps as it is one of their substrates. Thus, the comparison of A-C and G show the combination potency of either a BT or CCCP. Panels D-F and H show the penetration of EthBr into cells. The experimental results shown in Figure 7 are summarized in Table 3.

Each colored square represents the growth of bacteria in the presence of two compounds. Bacterial growth inhibition of greater than 95% is shown as white. For example, in Figure 7A, in the presence of half-MIC of 2 µM of W-*n*C_12_-W (**6**) and 0.5 µM norfloxacin, *S. aureus* growth was inhibited by >95%. A clear solution indicating no or very low growth of bacteria is also observed in the presence of 2 µM of compound **6** alone and 0.5 µM norfloxacin alone. At ½ MIC of W-*n*C_12_-W (**6**) (Figure 7A,D), complete recovery of norfloxacin and ethidium bromide potency was observed. Similar activities were observed with other compounds at ½ MIC (Figure 7B,C,E,F). The activity of the antibiotic was recovered regardless of resistance from efflux pumps in *S. aureus*. Several examples of such results are shown in Table 2.

Further analysis of combination activity was conducted by calculating the FIC Index (F-index) [49,50]. The F-index is defined as the (active concentration of BT/MIC of BT) + (active concentration of antibiotic/MIC of antibiotic). An F-index of <0.5 is considered to characterize synergistic activity. We observed synergistic activity with W-*m*C_6_H_4_-W (**1**) compounds (Figure 7C,F) at [norfloxacin] as low as 2 µM (F-index = 0.31). W-*n*C_12_-W (**6**) also showed synergy, but only with ethidium bromide, which is not used as an antibiotic. Overall, the three tryptophan compounds tested recovered antibiotic potency in resistant *S. aureus*. Of the three, W-*m*C_6_H_4_-W (**1**) showed the most promising combination activity despite its lower activity than **2** or **6**.

It is interesting to note that the relationship between membrane activity and antimicrobial potency for the BTs. The definition for synergy, the F-index of < 0.5, is somewhat arbitrary. However, a comparison of the *meta*-phenylene (**1**) and dodecylene (**6**) BTs, suggests that the former are membrane-active and effective ion transporters while the latter are membrane-active, but poor ion transporters (see Figure 2 and Figure 3 vs. Figure 4). The adjuvant studies of W-*n*C_12_-W (**6**) show additivity (F-index = 0.51), but not synergy with norfloxacin. In contrast, the F-index for *meta*-phenylene derivatives do show F-indices < 0.5. Our choice of *meta*-phenylene compounds (**1**, **2**) show synergy and ion transport ability, whereas the dodecylene compound (**6**) may be primarily a membrane disruptor. We therefore studied the *meta*-phenylene spacer in order to consider the ion transport and synergy functions.

### 2.4. Amino Acid Derivatives

Several of the previously prepared H_2_N-Trp-Y-Trp-NH_2_ derivatives showed antimicrobial activity [34]. The most active compound against *E. coli* was W-*n*C_12_-W (**6**). An obvious choice was therefore to prepare the range of amino acid derivatives as H_2_N-AAA-*n*C_12_-AAA-NH_2_ compounds. Three considerations directed us to choose W*-m*C_6_H_4_-W (**1**) as the model for our first pass analysis instead. First, isolation of W-*n*C_12_-W (**6**) had proved to be more difficult than for the arene derivatives, perhaps owing to the flexible spacer chain. Second, both ease of isolation for the *meta*-phenylene derivative and the ability to compare it to its structural isomers if activity was observed, favored W-*m*C_6_H_4_-W (**1**) as the spacer element. Third, the surprising difference in activity between the D,D- and L,L-isomers (**1** and **2**) would present an opportunity for further study if any other amino acid derivative proved to be active.

Thus, the compounds shown in Figure 8 were synthesized. All derivatives of *meta*-phenylenediamine. The terminal amino acids in each case have the L-stereochemistry and are alanine (**10**), leucine (**9**), lysine (**13**), phenylalanine (**7**), proline (**11**), threonine (**12**), tryptophan (**1**), and tyrosine (**8**). As noted above, the *bis*(tryptophan) derivative was previously prepared. The alanine (**10**) and leucine (**9**) derivatives were prepared to assess the influence of simple alkyl groups. Threonine and tyrosine possess hydroxyl groups having different pK_A_ values, but both can donate and accept hydrogen bonds. Tyrosine, as phenylalanine and tryptophan, is an electron-rich aromatic. The lysine derivative (**13**) doubles the number of positive charges. It was felt that this group of compounds represented a reasonable range of amino acid variations.

### 2.5. Synthesis and Characterization

The *t*-butyl carbamate-protected (Boc-protected) amino acids were coupled using (2-(1*H*-benzotriazol-1-yl)-1,1,3,3-tetramethyluronium hexafluorophosphate (HBTU) in DMF with diisopropylethylamine. After workup and isolation of the desired Boc-protected *bis*(amino)diamide product, deprotection was carried out using acid (HCl or trifluoroacetic acid) in CH_2_Cl_2_. Certain amino acid side chains required additional protecting groups. Their deprotection reactions are noted below and details may be found in the experimental section.

The side chain hydroxyl group of the *bis*(threonine) compound was protected as the benzyl ether, which was removed by catalytic hydrogenation prior to removal of the Boc group with trifluoracetic acid. The carbobenzyloxy (Cbz) group protected the *bis*(lysine) compound’s side-chain amines. The Cbz group was removed by catalytic hydrogenation prior to removal of the Boc group by treatment with trifluoroacetic acid to give the *tetrakis*(trifluoroacetate) salt. The overall synthetic approach is illustrated in Figure 1. Details of protection/deprotection sequences may be found in the experimental section.

### 2.6. Bacterial Strains Used

Two strains of bacteria were used for the studies reported here: Gram-negative *E. coli* and Gram-positive *S. aureus*. The strain of *E. coli* used was both tetracycline and ampicillin-resistant [34]. It was obtained by transforming competent JM-109 cells with the pBR322 plasmid. We have designated this strain as tet^R^ *E. coli*. Tetracycline resistance in this strain results from the expression of the TetA efflux pump. We have previously reported the antimicrobial potencies of several *bis*(tryptophan)s against tet^R^ *E. coli*. *S. aureus* 1199B is a clinically relevant strain that overexpresses the NorA efflux pump, but still shows a low level of resistance to norfloxacin. Combination studies with *bis*(amino acid)s and antibiotics were performed against both tet^R^ *E. coli* and *S. aureus* 1199B.

### 2.7. Antimicrobial Activity

We determined the minimum inhibitory concentrations (MICs) [51,52] of BTs against *S. aureus* 1199B. In previous work, [24] we studied *S. aureus* (ATCC 29213). Only the compounds that showed activity in our previous *S. aureus* screen were used for the study here. The data shown in Table 1, comport with values obtained for *S. aureus* 29213. The results for tet^R^ *E. coli* that are included in the table were identical to those obtained and published previously [34]. Compounds **2** and **6** showed the greatest activity. The most active compound had an alkyl spacer (W-*n*C_12_-W, **6**) and had MIC values of 4 µM and 10 µM against *S. aureus* 1199B and tet^R^ *E. coli*, respectively. Among the *bis*(tryptophan) compounds, *ortho*- (**3**), *meta*- (**1**, **2**) and *para*-phenylene (**4**) spacers, the D,D-*meta* (**2**) compound was most active against both organisms. The MIC of **2** was 16 µM against *S. aureus* 1199B and 28 µM against tet^R^ *E. coli*.

In preparing new structures **7**–**13**, the goal was to determine whether amino acids having side chain residues aside from the indole in tryptophan, the initial inspiration, would lead to antimicrobials. Alanine and leucine are hydrophobic amino acids and not expected to function as headgroups. Threonine and tyrosine have hydroxyl groups that are both polar and capable of hydrogen bonding. These polar elements could function as head groups. Tyrosine and phenylalanine are both electron-rich aromatics and could serve as head groups by dint of forming cation-pi interactions at membrane surfaces [53,54]. Proline is the only cyclic amino acid among the common 20 and no hypothesis was advanced about its amphiphilic behavior. Ultimately, none of **7**–**13** proved to exhibit antimicrobial properties.

### 2.8. In-Depth MIC Screen

Given that the most active compound in the BT family thus far studied incorporated the dodecylene spacer (i.e., W-*n*C_12_-W, **6**), we surveyed its activity more broadly than the two earlier examples: *E. coli* and *S. aureus*. The results of that survey are included in Table 3, above. In several cases, the potency study was arbitrarily limited to 18 µM (indicated as ‘>18′ µM).

Overall, the in-depth analysis showed that compound **6** (W-*n*C_12_-W) is more active towards Gram-positive than Gram-negative bacteria. The antimicrobial activity of **6** was, in most cases, inferior to that of commercial antibiotics. However, good activity (<10 µM) was observed for most of the Gram-positive bacteria such as MRSA, *E. faecium*, and *S. pneumoniae*. *S. pneumoniae* is a major cause of community-acquired pneumonia and MRSA is involved in various skin and soft tissue infections, hospital-acquired pneumonia, ventilator-associated pneumonia, etc.

Where vancomycin was tested against the indicated microbe, it was generally superior to the BTs. However, two cases show that the BTs are more potent and potentially applicable where drug resistance has developed. Vancomycin-resistant *Enterococcus faecalis* and *Enterococcus faecium* are sensitive to W-*n*C_12_-W. Meropenem resistant *Pseudomonas aeruginosa* and multidrug resistant *Klebsiella pneumoniae* are also sensitive to W-*n*C_12_-W. The low toxicity of W-*n*C_12_-W to mammalian cells and its potency both suggest W-*n*C_12_-W’s potential as a drug candidate.

## 3. Experimental Section

### 3.1. Planar Bilayer Conductance

Membranes were formed by painting lipid solutions (asolectin from sorbean dissolved in *n*-decane, 25 mg·mL^−1^, from Avanti Polar Lipids; Alabaster, AL, USA) over a 200 µM aperture separating two chambers containing 3 mL buffer solutions (450 mM KCl, 10 mM HEPES, pH = 7.00). The appropriate transporter (21 µL of a DMSO solution) was then added into the *cis* chamber (the side of the membrane that hosts the input electrode) to yield a final concentration of 7 µM. Working in a Faraday cage (Warner Instruments, Hamden, CT, USA) at room temperature, specific potentials were applied between two electrodes immersed in the two buffer solutions. The resulting currents were amplified (amplifier BC-525 D, from Warner Instruments), filtered with a 4-pole Bessel filter at 1 kHz, digitized by Digitizer (Digidata 1322A from Axon Instruments, Burlingame, CA, USA). The data were analyzed later using Clampfit 9.2 (Axon Instruments).

### 3.2. Fluorescence Assays

To test the membrane permeability of *S. aureus*, the bacteria were first grown in an incubator/shaker overnight from one CFU in media (37 °C; 200 RPM). *S. aureus* was then knocked back to O.D. = 0.550 (λ = 600 nm) before use. Cells were added to a sterile test tube followed by either of the following treatments: Triton X-100, DMSA or compounds **1**, **2**, and **6**. Compounds **1**, **2**, and **6** were added at concentrations of 0.5, 1, 2, 4, 8, 16 and 32 µM and incubated (37 °C; 200 RPM). The concentration of DMSO was kept constant at 0.5% *v/v* in each case. After incubation (30 min.) the cells were washed by centrifugation at 3000× *g* for 5 min and re-suspended in sterile phosphate-buffered saline (PBS Sigma Aldrich BioPerformance Certified). Propidium iodide (30 µM, Thermo-Fischer) was added to the *S. aureus* cells in PBS media, mixed by vortexing and incubated (37 °C; 200 RPM). After 30 min, the cells were loaded into a cuvette. Fluorescence data were collected by excitation at 493 nm and emission at 636 nm. The average readings from three separate experiments are reported herein in graphical format (Figure 6).

### 3.3. Determination of Minimum Inhibitory Concentrations (MICs)

The MIC values for each compound and organism combination were determined using standard protocols [24,38].

### 3.4. Compound Preparation

#### General Procedures

The preparation of compounds **1**–**6** has previously been reported [24].

Procedure 1. Coupling with HBTU

The *t*-butyl carbamate-protected (Boc-protected) amino acids and HBTU (2.1 equivalents) were dissolved in 10 mL anhydrous DMF with diisopropylethylamine (4.0 equivalents for the neutral diamines; 60 equivalents for diamine dihydrochlorides). The reaction was stirred overnight at room temperature under Argon. EtOAc (75 mL) was added and mixture was washed with 1 M NaHSO_4_ (2 × 75 mL), 5% NaHCO_3_ (3 × 50 mL), and brine. The organic layer was dried by filtration through a MgSO_4_/celite plug and the solvent removed in vacuo. The Boc-protected *bis*(amino acid) was used without further purification or was crystallized/precipitated from CH_2_Cl_2_/hexanes.

Procedure 2. Boc Deprotection with HCl/Dioxane

The Boc-protected *bis*(aminoamide) deprotection was carried out by using HCl (10 equiv) in dioxane/CH_3_OH and the product was obtained by precipitation and trituration with cold CH_2_Cl_2_.

Procedure 3. Boc Deprotection with TFA

Boc-protected *bis*(aminoamide) deprotection was carried out by using 10 or more equivalents of TFA neat or in CH_2_Cl_2_. The product was obtained by removal of the solvent and TFA in vacuo. The product was used without further purification or recrystallized from CH_3_OH/CHCl_3_/hexanes.

Procedure 4. Cbz Deprotection

The Cbz-protected amine was dissolved in absolute EtOH to which 0.1 molar equivalents of 10% Pd/C was added. The mixture was shaken on a Parr shaker at 60 psi for 4 h, then filtered through celite and the solvent was removed in vacuo.

Procedure 5. RO-*t*-Bu Deprotection with TFA

See general procedure 3.

Procedure 6. RO-CH_2_Ph (RO-Bn) Deprotection

See general procedure 4.


*Di-t-butyl ((2S,2′S)-(1,3-phenylenebis(azanediyl))bis(1-oxo-3-phenylpropane-2,1,diyl))dicarbamate (Boc-F-mPh-F-Boc)*


**7a** was prepared according to general procedure 1 using 1,3-phenylenediamine (0.150 mg, 1.39 mmol) and Boc-L-Phe-OH. White powder (**7a**, 0.80 g, 96% yield). ^1^H-NMR: 1.37 (s, 18H, (CH_3_)_3_), 2.90–3.20 (m, 4H, βCH_2_), 4.73 (m, 2H, αCH), 5.74 (br, 2H, Boc-NH), 6.92 (m, 1H, phenyl H5), 7.04 (m, 2H, phenyl H4, H6), 7.20 (m, 10H, C_6_H_5_), 7.61 (s, 1H, phenyl H2), 8.98 (br, 2H, PhNHCO-). ^13^C-NMR: 28.22, 38.57, 56.41, 80.14, 111.73, 115.63, 126.63, 128.39, 128.77, 129.17, 136.62, 137.90, 156.14, 170.43.


*(2S,2′S)-1,1′-(1,3-phenylenebis(azanediyl))bis(1-oxo-3-phenylpropan-2-aminium) chloride (F-mPh-F)*


**7**, was prepared according to general procedure 2 using **7a** (550 mg, 0.913 mmol). White powder (0.40 g, 93% yield). ^1^H-NMR (CD_3_OD): 3.14–3.34 (m, 4H, βCH_2_), 4.26 (t, *J* = 7.3 Hz, 2H, αCH), 7.26–7.35 (m, 13H, phenyl H4-6, C_6_H_5_), 7.90 (t, *J* = 1.7 Hz, 1H, phenyl H2). ^13^C-NMR (CD_3_OD): 38.76, 56.41, 113.14, 117.53, 128.88, 130.11, 130.33, 130.64, 135.53, 139.42, 167.93. Mass spectrum, calculated for free base C_24_H_26_N_4_O_2_. Found m/z 403.22 (M + 1).


*Di-t-butyl ((2S,2′S)-(1,3-phenylenebis(azanediyl))bis(3-(4-(benzyloxy)phenyl)-1-oxopropane-2,1-diyl))dicarbamate (Boc-Y(Bn)-mPh-Y(Bn)-Boc)*


**8a**, was prepared according to general procedure 1 using 1,3-phenylenediamine (0.150 mg, 1.39 mmol) and Boc-l-Tyr-(OBn)-OH. White powder (**8a**, 1.10 g, 97% yield). ^1^H-NMR: 1.38 (s, 18H, (CH_3_)_3_), 2.83–3.17 (m, 4H, βCH_2_), 4.70 (m, 2H, αCH), 4.94 (s, 4H, CH_2_Ph), 5.74 (br, 2H, Boc-NH), 6.84 (d, *J* = 8.4 Hz, 4H, phenolic H2, H6), 6.91 (m, 1H, phenyl H5), 7.04 (m, 2H, phenyl H4, H6), 7.12 (d, *J* = 8.4 Hz, 4H, phenolic H3, H5), 7.20–7.37 (m, 10H, C_6_H_5_), 7.67 (s, 1H, phenyl H2), 9.06 (br, 2H, PhNHCO-). ^13^C-NMR: 28.22, 37.69, 56.47, 69.72, 80.12, 111.61, 114.69, 115.52, 127.29, 127.72, 128.36, 128.76, 128.84, 130.20, 136.85, 137.93, 156.18, 157.53, 170.51.


*Di-t-butyl ((2S,2′S)-(1,3-phenylene-bis(azanediyl))bis(3-(4-hydroxyphenyl)-1-oxopropane-2,1-diyl))dicarbamate (Boc-Y-mPh-Y-Boc)*


**8a**, was prepared according to general procedure 6 using **8a** (1.00 g, 1.23 mmol). White solid (**8**, 0.77 g, 99% yield). ^1^H-NMR (2:1 CD_3_OD/CDCl_3_): 1.39 (s, 18H, (CH_3_)_3_), 2.82–3.11 (m, 4H, βCH_2_), 4.43 (m, 2H, αCH), 6.75 (d, *J* = 6.4 Hz, 4H, phenolic H2, H6), 7.05 (d, *J* = 6.5 Hz, 4H, phenolic H3, H5), 7.14–7.35 (m, 3H, phenyl H4-H6), 7.69 (s, 1H, phenyl H2), 9.64 (br, 2H, PhNHCO-). ^13^C-NMR: 28.68, 38.71, 57.66, 80.64, 113.29, 116.05, 117.30, 128.37, 129.77, 131.11, 139.07, 156.59, 157.03, 172.31.


*(2S,2′S)-1,1′-(1,3-phenylenebis(azanediyl))bis(3-(4-hydroxyphenyl)-1-oxopropan-2-aminium) chloride (Y-mPh-Y)*


**8**, was prepared according to general procedure 2 using **8a** (770 mg, 1.21 mmol). Off-white solid (0.61 g, 99% yield). ^1^H-NMR (CD_3_OD): 3.15–3.34 (m, 4H, βCH_2_), 3.70 (s, 2H, ArOH), 4.39 (m, 2H, αCH), 6.76 (m, 4H, phenolic H2, H6), 7.09–7.33 (m, 7H, phenyl H4-6, phenolic H3, H5), 8.09 (s, 1H, phenyl H2), 8.39 (br, 2H, PhNHCO-). ^13^C-NMR (CD_3_OD): 37.90, 56.78, 113.56, 116.93, 117.79, 126.13, 130.41, 132.04, 139.34, 158.03, 168.36. Mass spectrum, calculated for free base C_24_H_26_N_4_O_4_. Found m/z 435.22 (M + 1).


*Di-t-butyl ((2S,2′S)-(1,3-phenylenebis(azanediyl))bis(4-methyl-1-oxopentane-2,1-diyl))dicarbamate (Boc-L-mPh-L-Boc)*


**9a**, was prepared according to general procedure 1 using 1,3- phenylenediamine (0.150 mg, 1.39 mmol) and Boc-l-Leu-OH. White powder (0.66 g, 89% yield). ^1^H-NMR: 0.94 (dd, *J* = 6.9, 6.7 Hz, 12H, CH(CH_3_)_2_), 1.47 (s, 18H, (CH_3_)_3_), 1.55–1.85 (m, 6H, βCH_2_, γCH), 4.48 (m, 2H, αCH), 5.73 (m, 2H, Boc-NH), 6.85 (m, 1H, phenyl H5), 7.04 (m, 2H, phenyl H4, H6), 7.74 (s, 1H, phenyl H2), 9.30 (s, 2H, PhNHCO-). ^13^C-NMR: 21.42, 23.07, 24.61, 28.34, 41.15, 53.80, 80.02, 111.34, 115.03, 128.52, 156.42, 171.73.


*(2S,2′S)-1,1′-(1,3-phenylenebis(azanediyl))bis(4-methyl-1-oxopentan-2-aminium) chloride (L-mPh-L)*


**9**, was prepared according to general procedure 2 using **9a** (500 mg, 0.935 mmol). White powder (0.35 g, 91% yield). ^1^H-NMR (CD_3_OD): 1.04 (d, *J* = 5.8 Hz, 12H, CH(CH_3_)_2_), 1.71–1.85 (m, 6H, βCH_2_, γCH), 4.06 (t, *J* = 7.0 Hz, 2H, αCH), 7.29–7.34 (m, 1H, phenyl H5), 7.39–7.42 (m, 2H, phenyl H4, H6), 8.08 (t, J= 1.9 Hz, 1H, phenyl H2). ^13^C-NMR (CD_3_OD): 22.14, 23.28, 25.57, 41.77, 53.76, 113.11, 117.49, 130.49, 138.32, 139.73, 169.15. Mass spectrum, calculated for free base C_18_H_30_N_4_O_2_. Found m/z 335.25 (M + 1).


*Di-t-butyl ((2S,2′S)-(1,3-phenylenebis(azanediyl))bis(1-oxopropane-2,1-diyl))dicarbamate (Boc-A-mPh-A-Boc)*


**10a**, was prepared according to general procedure 1 using 1,3-phenylenediamine (0.100 mg, 0.92 mmol) and Boc-l-Ala-OH. White powder (0.35 g, 84% yield), mp 159–161 °C. ^1^H-NMR: 1.42 (d, J= 7.0 Hz, 6H, βCH_3_), 1.47 (s, 18H, (CH_3_)_3_), 4.37 (m, 2H, αCH), 5.31 (m, 2H, Boc-NH), 7.10 (m, 1H, phenyl H5), 7.19 (m, 2H, phenyl H4, H6), 7.60 (s, 1H, phenyl H2), 8.65 (s, 2H, PhNHCO-). ^13^C-NMR: 17.98, 28.35, 50.81, 80.46, 111.28, 115.50, 129.16, 138.22, 156.13, 171.39, 188.71.


*(2S,2′S)-1,1′-(1,3-phenylenebis(azanediyl))bis(1-oxopropan-2-aminium) chloride (A-mPh-A)*


**10**, was prepared according to general procedure 2 using **10a** (105 mg, 0.23 mmol). White powder (0.07 g, 93% yield), mp 195 °C (dec.). ^1^H-NMR (CD_3_OD): 1.60 (d, J= 7.0 Hz, 6H, βCH_3_), 4.08 (q, J= 7.0 Hz, 2H, αCH), 7.22–7.39 (m, 3H, phenyl H4, H5, H6), 8.05 (m, 1H, phenyl H2). ^13^C-NMR (CD_3_OD): 17.89, 51.06, 112.94, 117.34, 130.50,139.86,169.48. Mass spectrum, calculated for free base C_12_H_18_N_4_O_2_. Found m/z 251.16 (M + 1).


*Di-t-butyl-(2S,2′S)-2,2′-((1,3-phenylenebis(azanediyl))bis(carbonyl))bis-(pyrrolidine-1-carboxylate) (Boc-P-mPh-P-Boc)*


**11a**, was prepared according to general procedure 1 using 1,3-phenylenediamine (0.150 mg, 1.39 mmol) and Boc-l-Pro-OH. White powder (0.57 g, 81% yield), mp 121 °C (dec.). ^1^H-NMR: 1.49 (s, 18H, (CH_3_)_3_), 1.85–2.55 (m, 8H, βCH_2_, γCH_2_), 3.31–3.67 (m, 4H, δCH_2_) 4.30–4.48 (m, 2H, αCH), 7.06–7.52 (m, 3H, phenyl H4-6), 7.62 (br, 1H, phenyl H2), 9.34 (br, 2H, PhNHCO-). ^13^C-NMR: 24.58, 28.47, 47.28, 60.71, 80.83, 110.91, 115.37, 129.23, 138.71, 156.32, 170.35.


*(2S,2′S)-2,2′-((1,3-phenylenebis(azanediyl))bis(carbonyl))bis(pyrrolidin-1-ium) 2,2,2-trifluoroacetate (P-mPh-P)*


**11**, was prepared according to general procedure 3 using **11a** (400 mg, 0.80 mmol). White powder (0.39 g, 91% yield), mp 80 °C (dec.). ^1^H-NMR (DMSO-d_6_): 1.87–2.04 (m, 6H, γCH_2_, βCH), 2.32–2.43 (m, 2H, βCH), 3.30 (m, 4H, δCH_2_), 4.41 (m, 2H, αCH), 7.35 (m, 3H, phenyl H4-6), 8.07 (s, 1H, phenyl H2). 8.79 (br, 2H, NH_2_^+^), 9.90 (br, 2H, NH_2_^+^), 10.77 (s, 2H, PhNHCO-). ^13^C- NMR (DMSO-d_6_): 23.52, 29.65, 45.69, 59.63, 110.69, 115.16, 129.35, 138.59, 158.65, 166.93. Mass spectrum, calculated for free base C_16_H_22_N_4_O_2_. Found m/z 303.19 (M + 1).


*Di-t-butyl ((2S,2′S,3R,3′R)-(1,3-phenylene-bis(azanediyl))bis(3-(benzyloxy)-1-oxobutane-2,1-diyl))dicarbamate (Boc-T(Bn)-mPh-T(Bn)-Boc)*


**12a**, was prepared according to general procedure 1 using 1,3-phenylenediamine (150 mg, 1.39 mmol) and Boc-l-Thr-(OBn)-OH. White powder (0.90 g, 94% yield). ^1^H-NMR: 1.21 (d, J: 6.4 Hz, 6H, γCH_3_), 1.44 (s, 18H, (CH_3_)_3_), 4.15 (m, 2H, BCH), 4.44–4.61 (m, 6H, αCH, OCH_2_Ph), 5.83 (d, *J* = 6.8 Hz, 2H, Boc-NH), 7.05–7.31 (m, 13H, OBn, phenyl H4-6), 7.74 (s, 1H, phenyl H2), 8.80 (br, 2H, PhNHCO-). ^13^C-NMR: 15.35, 27.84, 58.16, 70.89, 74.43, 79.58, 111.07, 115.36, 127.28, 127.83, 127.98, 128.78, 137.39, 137.69, 155.63, 168.18.


*(2S,2′S,3R,3′R)-1,1′-(1,3-phenylenebis(azanediyl))bis(3-hydroxy-1-oxobutan-2-aminium) 2,2,2-trifluoroacetate (T-mPh-T)*


**12**, was prepared by benzyl deprotection from **12a** (0.9 g, 1.30 mmol) according to general procedure 6 (0.66 g, 99% yield) followed by Boc deprotection by general procedure 3. Off-white solid (0.36 g, 93% yield), mp 85 °C (dec.). ^1^H-NMR (CD_3_OD): 1.33 (d, *J* = 6.4 Hz, 6H, γCH_3_), 3.81 (d, *J* = 6.2 Hz, 2H, αCH), 4.14 (m, 2H, βCH), 7.20–7.40 (m, 3H, phenyl H4-6), 8.07 (s, 1H, phenylH2). ^13^C-NMR (CD_3_OD): 20.38, 60.99, 67.58, 113.04, 117.50, 130.52, 139.65, 167.03. Mass spectrum, calculated for free base C_14_H_22_N_4_O_4_. Found m/z 311.16 (M + 1).


*Di-t-butyl((2S,2′S)-(1,3-phenylenebis(azanediyl))bis(6-benzylcarboxyamido-1-oxohexane-2,1-diyl))dicarbamate (Boc-K(Cbz)-mPh-K(Cbz)-Boc)*


**13a**, was prepared according to general procedure 1 using 1,3-phenylenediamine (150 mg, 1.39 mmol) and Boc-l-Lys-(NεCbz)-OH. Yellow solid (1.10 g, 95% yield). ^1^H-NMR: 1.33–1.55 (m, 26H, (CH_3_)_3_, γCH_2_, δCH_2_), 1.58–1.86 (m, 4H, βCH_2_), 3.13 (m, 4H, δCH_2_), 4.29 (m, 2H, αCH_2_), 5.06 (s, 4H, CH_2_Ph), 5.59 (br, 2H Cbz-NH), 5.92 (br, 2H, Boc-NH), 7.06 (m, 1H, phenyl H5), 7.23 (m, 12H, Cbz-ArH, phenyl H4, H6), 7.99 (s, 1H, phenyl H2), 9.18 (br, 2H PhNHCO-).


*(5S,5′S)-6,6′-(1,3-phenylenebis(azanediyl))bis(6-oxohexane-1,5-diaminium) 2,2,2-trifluoroacetate*


**13** was prepared according to general procedure 4 using **13a** (1.10 g, 1.32 mmol). Cbz deprotected product **13b** (0.60 g, 81% yield) was used without further purification. Boc deprotection by general procedure 3 using **13b** (600 mg, 1.06 mmol) afforded K-*m*Ph-K, **13**, as a yellow solid (0.76 g, 87% yield). ^1^H-NMR (CD_3_OD): 1.58 (m, 4H, γCH_2_), 1.75 (m, 4H, αCH_2_), 2.01 (m, 4H, βCH_2_), 2.98 (t, *J* = 7.6 Hz, 4H, εCH_2_), 4.09 (t, *J* = 6.4 Hz, 2H, αCH), 7.28–7.40 (m, 3H, phenyl H4-6), 8.08 (s, 1H, phenyl H2). ^13^C-NMR (CD_3_OD): 23.05, 28.20, 32.29, 40.35, 54.96, 113.26, 117.62, 130.54, 139.70, 162.95, 163.49, 168.67. Mass spectrum, calculated for free base C_18_H_32_N_6_O_2_. Found m/z 365.25 (M + 1).

## 4. Conclusions

Herein we report the synthesis of compounds having the general formula Aaa-linker-Aaa where the amino acids were chosen from among the common 20. Derivatives other than Aaa = Trp, i.e., the *bis*(tryptophan) compounds, showed no activity as antimicrobials. This highlights the importance of tryptophan and the novelty of the *bis*(tryptophan) compound family developed, further supporting the hypothesized mechanism. To summarize, we report three novel findings. First, a range of bacteria succumb to the activity of various *bis*(tryptophan) compounds. W-*n*C_12_-W (**6**) shows a MIC value of 4 µM against *S. aureus* 1199B and MIC values of 9 µM and 18 µM respectively against *E. faecium* VanA and *E. faecalis* Van B, which are resistant to multiple antimicrobials, including vancomycin. Second, evidence has been obtained that clearly shows the formation of pores by BTs in planar bilayers. The formation of ion-conducting pores in bilayers does not guarantee identical behavior in Gram negative or Gram positive membranes. However, confirmation of channel formation is consistent with the hypothesis that membrane permeability is enhanced by these ionophores and that the disruption of ion balance as a result may hinder efflux pump function as well as other ion-regulated enzymatic processes. Fluorescence studies also support the hypothesis of enhanced membrane penetration.

Third, the low toxicity *bis*(tryptophan) compounds having either *meta*-phenylene or *n*-dodecylene linkers function as effective adjuvants to enhance the properties of FDA approved antimicrobials against organisms such as *S. aureus*. An example reported herein shows a BT-mediated synergistic effect that enhances the potency of norfloxacin against *S. aureus* by 128-fold.

Finally, W-*n*C_12_-W is both a membrane disruptor and an antibiotic *per se* as evidence by the propidium iodide experiments (Figure 6), the planar bilayer study, and its low MIC as shown in the in-depth screen. The *meta*-phenylene compounds are effective adjuvants as they form pores (Figure 2) and show synergistic activity (F-index < 0.5).

## Data Availability

Data are contained within the article.

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
