# Peer review of "Bis(Tryptophan) Amphiphiles Form Ion Conducting Pores and Enhance Antimicrobial Activity against Resistant Bacteria"

_antibiotics, 2021, doi:10.3390/antibiotics10111391_

Round 1

Reviewer 1 Report

In the present manuscript, the authors examined bis(tryptophan) amphiphiles (compounds 1-6), which were prepared in a previous study and exhibited antibacterial properties, for their ability to insert into bacterial membranes and form channels, thereby increasing membrane permeability. For some of the compounds, channel formation (compounds 2 and 3) and increased membrane penetration (compounds 2 and 6) were confirmed. In addition, the compounds with the highest potency (1, 2 and 6) were screened for antimicrobial synergy, with compounds 1 and 2 showing synergy. Moreover, derivatives (compounds 7-13) of compound 1 containing different side chain residues were prepared to determine their antibacterial potency but were unfortunately inactive. Finally, the most active compound (compound 6) was tested against an extended range of bacterial strains, indicating high antibacterial potency against some multidrug-resistant bacteria.

The manuscript is generally well structured and presented, with valid and reasoned conclusions. However, the following suggestions may help to further improve the manuscript.

The BLM experiment was used to detect membrane insertion of compounds and it is unclear whether this experiment was performed with all 6 compounds or only compounds 1-4, as the results are only given and explained for compounds 1-4. It is not clear how the authors conclude that compound 6 is a membrane disruptor without investigating its effect on the membrane. In my opinion, it would be very appropriate if the results were also presented for compounds 5 and 6 to see what kind of effect they have on the membrane, especially since compound 6 is the most active. Please include the results, or at least explain why it was not done and explain the basis for your conclusion that compound 6 is a membrane disruptor.

In the membrane permeability experiment, Triton X-100 and in the checkerboard experiment, CCCP were used as controls. It is not clear why these two compounds were chosen as controls. I think their role should be explained in the manuscript.

In the manuscript, the compounds are sometimes referred to by their compound number (1-13), but most of the time they are referred to by the compound name (for example, W-nC12-W). I found it quite hard to follow the text, especially when only the name was used. So, I suggest that the compounds are also referred to by the compound number each time.

The preparation of the new compounds 7-13 was described in this manuscript and the structures were confirmed by 1H and 13C NMR. To sufficiently confirm the structures, at least high-resolution mass spectrometry or mass spectrometry should be performed.

Line 109: "As recorded below,..." Please indicate where these results are shown.

Line 154: "The conductances observed for compounds that formed stable pores..." Please indicate which compounds were able to form stable pores.

Line 174: "The data in Table 1 (presented later in this report)..." I believe this should be Table 2, not Table 1.

Lines 198-202: Please indicate where and in which table the results of the MICs are presented.

Lines 225-226: "In previous studies with synthetic amphiphiles, we found that synergy with FDA 225 approved drugs was possible." Please provide the reference to these previous studies.

Line 282: The author mentions Figure 4, but there is no Figure 4 in the manuscript.

Author Response

In the present manuscript, the authors examined bis(tryptophan) amphiphiles (compounds 1-6), which were prepared in a previous study and exhibited antibacterial properties, for their ability to insert into bacterial membranes and form channels, thereby increasing membrane permeability. For some of the compounds, channel formation (compounds 2 and 3) and increased membrane penetration (compounds 2 and 6) were confirmed. In addition, the compounds with the highest potency (1, 2 and 6) were screened for antimicrobial synergy, with compounds 1 and 2 showing synergy. Moreover, derivatives (compounds 7-13) of compound 1 containing different side chain residues were prepared to determine their antibacterial potency but were unfortunately inactive. Finally, the most active compound (compound 6) was tested against an extended range of bacterial strains, indicating high antibacterial potency against some multidrug-resistant bacteria.

The manuscript is generally well structured and presented, with valid and reasoned conclusions. However, the following suggestions may help to further improve the manuscript.

The BLM experiment was used to detect membrane insertion of compounds and it is unclear whether this experiment was performed with all 6 compounds or only compounds 1-4, as the results are only given and explained for compounds 1-4.

It is not clear how the authors conclude that compound 6 is a membrane disruptor without investigating its effect on the membrane. In my opinion, it would be very appropriate if the results were also presented for compounds 5 and 6 to see what kind of effect they have on the membrane, especially since compound 6 is the most active. Please include the results, or at least explain why it was not done and explain the basis for your conclusion that compound 6 is a membrane disruptor.

I have included the trace and the appropriate discussion.

In the membrane permeability experiment, Triton X-100 and in the checkerboard experiment, CCCP were used as controls. It is not clear why these two compounds were chosen as controls. I think their role should be explained in the manuscript.

Triton X-100 is a detergent and is used as a control in the membrane permeability experiment because it disrupts biological membranes. In the checkerboard experiments, CCCP is an efflux pump inhibitor and Triton X-100 is a detergent control. We have included this information in the text.

In the manuscript, the compounds are sometimes referred to by their compound number (1-13), but most of the time they are referred to by the compound name (for example, W-nC12-W). I found it quite hard to follow the text, especially when only the name was used. So, I suggest that the compounds are also referred to by the compound number each time.

I believe that all occurrences of name abbreviations are now accompanied by a numerical designation.

The preparation of the new compounds 7-13 was described in this manuscript and the structures were confirmed by 1H and 13C NMR. To sufficiently confirm the structures, at least high-resolution mass spectrometry or mass spectrometry should be performed.

Mass spectrometric results have been added for compounds 7-13.

Line 109: "As recorded below,..." Please indicate where these results are shown.

Now Table 2 is specified.

Line 154: "The conductances observed for compounds that formed stable pores..." Please indicate which compounds were able to form stable pores.

I have changed this sentence to to read “The conductances observed for compounds 2 and 3, which formed stable pores,...”

Line 174: "The data in Table 1 (presented later in this report)..." I believe this should be Table 2, not Table 1.

Corrected. Thank you.

Lines 198-202: Please indicate where and in which table the results of the MICs are presented.

Tables 2 and 3

Lines 225-226: "In previous studies with synthetic amphiphiles, we found that synergy with FDA 225 approved drugs was possible." Please provide the reference to these previous studies.

I have cited reference 23a to which this statement refers.

Line 282: The author mentions Figure 4, but there is no Figure 4 in the manuscript.

The Reviewer is correct. Figure 4 has been added and the text altered appropriately. 

Reviewer 2 Report

The manuscript entitled "Bis(Tryptophan) Amphiphiles Form Ion Conducting Pores and  Enhance Antimicrobial Activity Against Resistant Bacteria" by M. Patel et al. presents synthesis of new compounds 7-13, and antibacterial activity of compounds obtained earlier (1-6) and newly synthesized (7-13). For active compounds, Authors also performed other interesting studies in order to explain the mechanism of their action.

The manuscript needs an improvement:

  1. The manuscript should be rewritten. The Authors do not present results and discussion in a logical order e.g. the checkerboard experiments, which are based on MIC values, are presented before evaluation of antibacterial activity.
  2. The section "2. Results" should be renamed to "2. Results and discussion".
  3. Figure 1 and 8 have poor quality, they should be improved.
  4. A scheme of synthesis should be added to the manuscript. The Authors introduced compound designations 7a-13a without their chemical structures.
  5. The spectral characteristics of newly synthesized compounds should be supplemented with mass spectra.
  6. It is not clear, what compound Authors describe in lines 548-556.

Author Response

The manuscript should be rewritten. The Authors do not present results and discussion in a logical order e.g. the checkerboard experiments, which are based on MIC values, are presented before evaluation of antibacterial activity.

I must respectfully disagree with this criticism. Reviewer 1 stated “The manuscript is generally well structured and presented, with valid and reasoned conclusions. However, the following suggestions may help to further improve the manuscript.” We have made the corrections/followed the suggestions of Reviewer 1 and do not believe that the order of presentation is illogical. Of course, we have rewritten the manuscript in several places and believe that the manuscript is improved as a result of reviewer suggestions. A problem is that the data in the checkerboard section and the MIC data are related. One of the section must come first. I think the organization in this case is appropriate.

The section "2. Results" should be renamed to "2. Results and discussion".

The section has been renamed.

Figure 1 and 8 have poor quality, they should be improved.

Both figures have been redrawn at a higher resolution. The figure has also been corrected to show terminii of NH3Cl rather than NHCl.

A scheme of synthesis should be added to the manuscript. The Authors introduced compound designations 7a-13a without their chemical structures.

The synthetic approaches for 7a-13a are essentially the same. We have provided a general synthetic scheme (“Scheme 1”) at the nd of section 2.5. The pendant residue, designated R, will be obvious from the structures shown as 7-13 in Figure 8.

The spectral characteristics of newly synthesized compounds should be supplemented with mass spectra.

These data have been included as explained above.

It is not clear, what compound Authors describe in lines 548-556.

The compound number 6 has been changed to 13, to which the procedure refers. The procedure numbers have also been changed to reflect the subsection numbers used in the journal.

Round 2

Reviewer 2 Report

The manuscript has been corrected according to Reviewers' suggestions and can be accepted to publication.